# Application of Unconventional Tillage Systems to Maize Cultivation and Measures for Rational Use of Agricultural Lands

Felicia Chețan [1], Teodor Rusu [2,*], Cornel Chețan [1], Alina Șimon [1], Ana-Maria Vălean [1,*], Adrian Ovidiu Ceclan [1], Marius Bărdaș [1] and Adina Tărău [1]

[1]  Agricultural Research and Development Station Turda, Agriculturii Street 27, 401100 Turda, Romania; felicia.chetan@scdaturda.ro (F.C.); cornel.chetan@scdaturda.ro (C.C.); alina.simon@scdaturda.ro (A.Ș.); adrian.ceclan@scdaturda.ro (A.O.C.); marius.bardas@scdaturda.ro (M.B.); adina.tarau@scdaturda.ro (A.T.)

[2]  Department of Technical and Soil Sciences, Faculty of Agriculture, University of Agricultural Sciences and Veterinary Medicine Cluj-Napoca, Mănăstur Street 3–5, 400372 Cluj-Napoca, Romania

*   Correspondence: trusu@usamvcluj.ro (T.R.); anamaria.valean@scdaturda.ro (A.-M.V.)

**Abstract:** Maize (*Zea mays* L.) is one of the main agricultural crops grown worldwide under very diverse climate and soil conditions. For maize cultivation in a conventional tillage system, autumn plowing is a mandatory condition. Minimum soil tillage or no tillage has been applied in recent years, both in research and in production, for reasons relating to soil conservation and fuel economy. This paper presents the results of the research executed under pedoclimatic conditions at the Agricultural Research and Development Station Turda (ARDS Turda, Romania; chernozem soil) regarding the behavior of the maize hybrid Turda 332 cultivated in four tillage systems and two levels of fertilization during the period of 2016–2022. The following soil tillage systems were applied: a conventional tillage system (CT) and unconventional tillage systems in three variants—a minimum tillage system with a chisel (MTC), a minimum tillage system with a disk (MTD), and a no-tillage system (NT). They were applied with two levels of fertilization: basic fertilization (350 kg ha$^{-1}$ NPK 16:16:16, applied at sowing) and optimized fertilization (350 kg ha$^{-1}$ NPK 16:16:16 applied at sowing + 150 kg ha$^{-1}$ calcium ammonium nitrate with additional fertilization in the phenophase of the maize with 6–7 leaves). The results highlight the fact that under the conditions of chernozem soils with a high clay content (41% clay content), maize does not lend itself to cultivation in MTD and NT, requiring deeper mobilization, with the yield data confirming this fact. This is because under the agrotechnical conditions for sowing carried out in MTD and NT, the seeder used (Maschio Gaspardo MT 6R) does not allow for the high-quality sowing of maize, especially under dry soil conditions. Instead, the MTC system could be an alternative to the conventional tillage system, with the yield difference being below 100 kg ha$^{-1}$.

**Keywords:** minimum tillage; no tillage; fertilization; chernozem; maize; yield

## 1. Introduction

Maize (*Zea mays* L.) represents an important crop with multiple uses and advantages [1–3]. Being a very versatile crop, it is used in human nutrition [4] as an important source of minerals, fiber, carbohydrates, and a complex of vitamins A and B; as fodder grains, flour, silage, and cob; in the production of alcohol, starch, dextrin, glucose, and corn oil; and in obtaining bioethanol, bio-gas, etc. [5]. It is cultivated around the globe under very diverse climate and soil conditions [6], found in the northern hemisphere up to 58° (Canada and Russia), in the southern hemisphere up to 42–43° (New Zealand), and between 42° south latitude and 53° northern latitude [7]. Along with wheat and rice, maize provides over 30% of food calories, and is a source of protein in over 90 countries [8–10]. Maize is of major economic importance not only due to its high production potential but also because

of the diversity of products obtained by cultivating it. However, the soil tillage system plays an important role in the production potential [10]. Worldwide, there are four major maize-producing regions, of which North America is in first place, representing over 30% of global production; followed by China (over 20% of global production); Europe (12% of global production); and South America, which produces 15%, mostly in the territories of Brazil, Argentina, and Mexico [11]. Regarding exports, the United States of America occupies the first place by far, while Romania ranks in sixth place, representing 3.3% of the world exports of this product [10]. Worldwide, it is estimated that unconventional soil tillage systems are practiced on an area of more than 70 million ha [4], most of which is in Latin America, the United States of America, and Australia, with only a small part in other areas of the world.

In Romania, half of the total area cultivated with cereals is occupied by maize; therefore, maize holds one of the first positions in the country's cereal economy [12]. Compared to other areas of Romania, the Transylvanian Plain has some specific characteristics that can create problems in the cultivation of maize: a deficient rainfall regime, especially in July; a surplus thermal regime and a relatively shorter frost-free interval; climatic diversity; a rugged relief; and soils often with different particularities, even from one plot to another [13]. Knowledge of the scientific developments in agriculture allows for the choice of the latest technologies [14–17] to meet the requirements of producers and consumers alike [18].

Soil management systems need to be developed to address the emerging issues of the 21st century: global climate change, accelerated soil degradation and desertification, the decline in biodiversity, and the achievement of food security [19,20]. The choice of soil tillage system must take into account the available equipment, the type of soil, and the climatic conditions [21]. According to some authors, conventional tillage systems are the main reason for accelerated soil erosion [22,23]. Plowing, as well as frequent harrowing, results in the depletion of soil nutrients, such as organic carbon, nitrogen, phosphorus, and potassium [24].

Research on different soil tillage systems is very popular worldwide, and these systems are increasingly applied by farmers to different crops [15]. The use of a soil tillage system is an essential maize-growing practice for successful production. These systems can significantly influence the yield and nutritional quality of maize via their effects on soil and moisture conservation, temperature, aeration, nutrient availability, and cost and labor savings [15,25].

In research carried out in some regions with a temperate climate, maize yields under no tillage were found to be similar to or lower than those of the conventional tillage system [26,27]. The conventional tillage system allows for a more pronounced mobilization of the soil, oxidation of the organic matter in the soil, and, thus, potentiation of soil fertility. Over time, this action on the soil can lead to soil depletion if we do not intervene with compensatory amounts of organic matter [28]. Minimum tillage and no-tillage systems reduce inputs, making these systems more attractive to farmers. However, these tillage systems do not succeed without suitable equipment, without mulch on the soil surface, and without additional knowledge of their application [16].

It is not only the type of soil but also the soil tillage system and used agricultural machinery that have a major effect on maize yield [16]. In some studies carried out in a region with a temperate climate, it turned out that the maize yields of the NT system were lower than those of CT, but, instead, they had the advantages of better retaining water and increasing the rate of water infiltration, contributing to reducing the input of work and fuel [29,30] after several years of application.

In a long-term experiment in the period of 2005–2016, Simić et al. [31] showed that the grain yield of maize was 10.0, 8.3, and 7.0 t ha$^{-1}$ under CT, reduced tillage (RT), and NT, respectively, while in dry years, the maize grain yield was higher under reduced tillage than CT. Numerous studies carried out by other authors also showed the effects of soil tillage systems on maize production and its components [32,33].

The mechanization of agricultural practices must be adapted to meet soil protection requirements, and soil and water conservation practices are necessary in many areas due to the soil degradation caused by using only conventional tillage systems for many years [34–36]. It is difficult to predict the reaction of the maize crop to the soil tillage system, as the yields are influenced by several factors, such as soil characteristics, the microclimate [37], and the effects of different practices (the degree of soil mobilization, sowing, the equipment used, crop rotation, hybrids, fertilization, weed control, etc. [38–43]).

Some studies have shown that unconventional tillage systems are better applied to soil with a high content of organic matter, loamy-textured soils, and well-drained soils [44–46]. Research carried out on the chernozem soil type has shown that maize yields are lower in NT and RT than in CT when either crop rotation or continuous maize production is applied [15,47,48]. The hypothesis of the research carried out by us is related to determining the effect of the combination of the soil tillage system with fertilization and the impact of the climatic conditions of the year (taken as an experimental factor).

In this context, the aim of this research is to find a balance between the soil tillage systems and the produced effects on maize yield. In this paper, the results of the research executed under pedoclimatic conditions at the Agricultural Research and Development Station Turda (ARDS Turda, Romania) are presented regarding the behavior of the maize hybrid T 332 cultivated in four tillage systems (a conventional tillage system; minimum tillage with a chisel; minimum tillage with a disk; and no tillage) and two levels of fertilization, during the period of 2016–2022.

## 2. Materials and Methods

### 2.1. Experimental Site

This research was carried out in an agroecosystem in the Transylvanian Plateau. The Transylvanian Plateau is described as a plateau bordered to the south, east, and northeast by the Carpathian Mountains and to the west by the Apuseni Mountains. It consists of hills with heights between 500 and 600 m and wide or short valleys. The existence of valleys and extensive depressions facilitates the easy penetration of air masses from neighboring regions, promoting the occurrence of weather changes and temperature inversions.

The Agricultural Research and Development Station Turda (ARDS Turda), from a geographical point of view, is located in the northwestern part of the Turda municipality (Figures 1 and 2) at a distance of 30 km from Cluj-Napoca and 3 km from the most important traffic artery of Romania DN1-E60-E81. The geographical coordinates of the research station are 46° and 35′ north latitude and 23° and 47′ east longitude Greenwich at an altitude of 345–493 m from the Adriatic Sea.

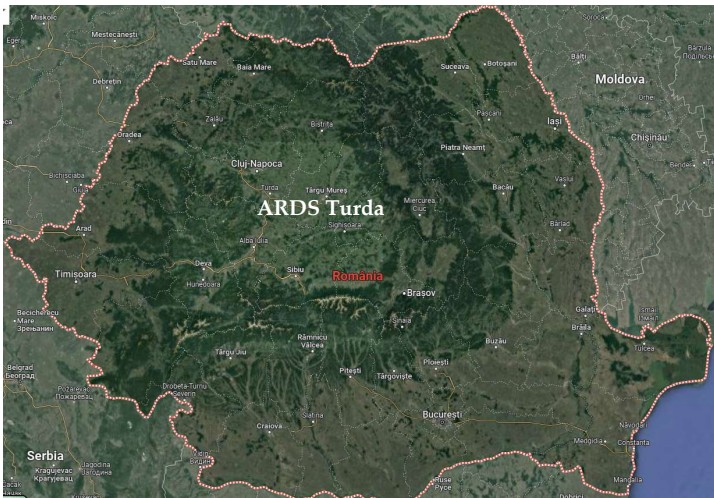

**Figure 1.** Location of ARDS Turda.

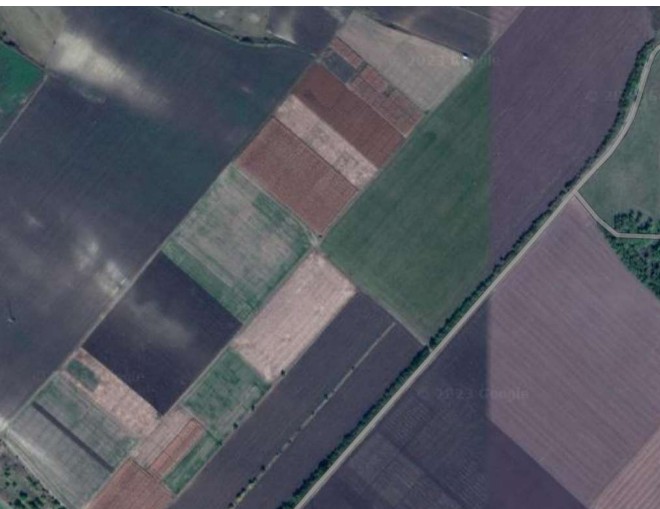

**Figure 2.** Experimental fields of ARDS Turda.

The general climate of the area is characterized by the values recorded at the Turda Weather Station, defined as a boreal climate with continental characteristics. Ever since 1990, there has been a process of increased warming in our country, and the same thing is happening in Transylvania. Under the conditions of Turda in 1982, the multiannual average temperature was 8.4 °C; in 2007, it increased to 8.9 °C, which could lead to an increase of 2 °C in the next 25 years [49].

The thermal regime is characterized by an annual air average of 9.2 °C, where the warmest month is July with an average monthly temperature of 19.7 °C and the coldest month is January with an average monthly temperature of −3.4 °C. The absolute minimum temperature was −36.5 °C, recorded in the winter of 1963, and the absolute maximum temperature was 38.5 °C, recorded in the summer of 1946 [50].

### 2.2. Collection and Recording of Weather and Rainfall Data

The temperature and precipitation presented in this paper come from the Turda Weather Station, which is located near the experimental fields. The Turda Weather Station is a subunit of the Northern Transylvania Regional Meteorological Center, part of the Romanian National Meteorological Administration.

### 2.3. Biological Materials

The biological material used in the experimental study is the simple maize hybrid T 332, creation ARDS Turda, (Figure 3), which was registered in the Official Catalog of Crop Varieties from Romania in 2015 and included in the maturity group FAO 380 [51]. The hybrid presents good tolerance to the low temperatures recorded in the first part of the vegetation period, tolerates periods of drought quite well, and has a medium resistance to the attack of the pest *Ostrinia nubilalis* [52,53].

### 2.4. Experimental Design

The present paper presents the results of the research executed under the pedoclimatic conditions at ARDS Turda, tracking the yield of maize under the influence of four variants of tillage and two variants of fertilization during the period of 2016–2022.

Experimental factors:

A. Experimental year: $a_1$, 2016; $a_2$, 2017; $a_3$, 2018; $a_4$, 2019; $a_5$, 2020; $a_6$, 2021; and $a_7$, 2022.

B. Soil tillage system: $b_1$, a conventional tillage system (CT) with plowing (30 cm depth); $b_2$, minimum soil tillage with a chisel (MTC, 30 cm depth); $b_3$, minimum soil tillage with a disk (MTD, 15 cm depth); and $b_4$, no-tillage (NT, direct sowing).

C.  Level of fertilization: $c_1$, 350 kg ha$^{-1}$ NPK 16:16:16 at sowing and $c_2$, 350 kg ha$^{-1}$ NPK 16:16:16 at sowing + 150 kg ha$^{-1}$ calcium ammonium nitrate (CAN) that contains roughly 8% calcium and 27% nitrogen.

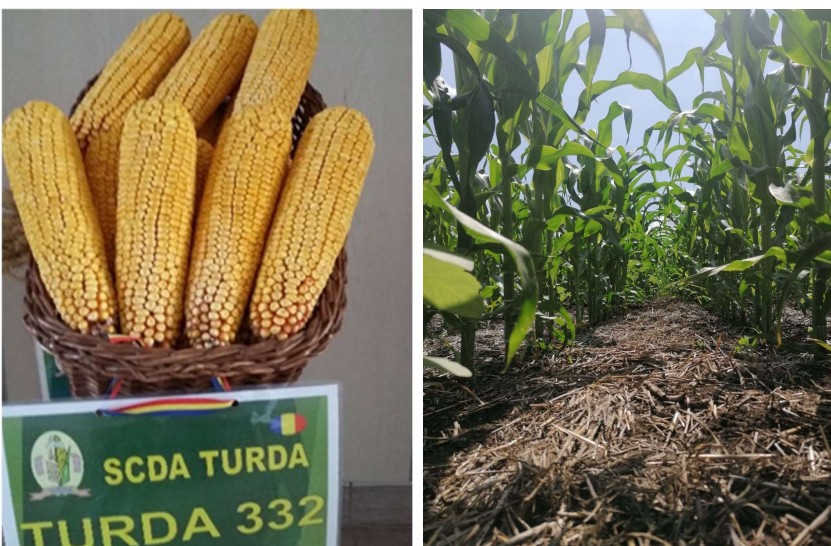

**Figure 3.** Turda 332 maize hybrid.

### 2.5. Technology Used at the Experimental Site

The crops are rotated every three years (soybean, winter wheat, and maize); this rotation is the most frequently used in Romania. The experimental site was located on a clay-illuvial chernozem-type soil [54], with a clay–clay texture (41% clay content), neutral pH (6.9), 2.95% humus content, 0.211% total nitrogen, 23 ppm phosphorus, and 283 ppm potassium [55]. The experimental design used a split-plot method arranged in three repetitions. The area of an experimental plot was 48 m$^2$ (4 m width and 12 m length).

The sequence of works performed according to the experimental variants is presented in Table 1.

**Table 1.** The sequence of works performed in the experimental field.

| No. | Work Period | Soil Tillage | Agrotechnical Work | Aggregate Used |
|---|---|---|---|---|
| 1. | The last 10 days of October | CS | Plowing (30 cm depth) | Plow Kuhn Multi-Master 125T + tractor John Deere 6620SE |
| 2. | The last 10 days of October | MTC | Chisel (30 cm depth) | Chisel Pinocchio Gaspardo + tractor John Deere 6620SE |
| 3 | The last 10 days of October | MTD | Disking (15 cm deep) | Disk GDU Gaspardo 3.4 + tractor John Deere 6620SE |
| 4. | The first 10 days of April | CS, MTC, MTD | Working with rotary harrow | Rotary Harrow Kuhn HRB 403 D + tractor John Deere 6620SE |
|  |  | CS, MTC, MTD, NT | Sowing + fertilized I | Seeder Maschio Gaspardo MT 6R + tractor John Deere 6620 SE |
|  |  | CS, MTC, MTD, NT | Pre-emergent weed control on the soil | Herbicide machine MET 1500 + tractor John Deere 6620 SE |
| 5. | The last 10 days of May | CS, MTC, MTD, NT | Weed control on vegetation | Herbicide machine MET 1500 + tractor John Deere 6620 SE |
| 6. | The first 10 days of June | CS, MTC, MTD, NT | Fertilized II | Gaspardo Zeno + tractor John Deere 6620 SE |
| 7. | October | CS, MTC, MTD, NT | Harvest | Experimental combine Wintersteiger (Wintersteiger AG, Austria) |

The sowing rate was 65,000 plants ha$^{-1}$, the seeds were treated with the fungicide Maxim XL 035 FS (1.0 L t$^{-1}$ product based on fludioxonil 25 g L$^{-1}$ + metalaxyl − M (mefenoxam) 10 g L$^{-1}$) before sowing, and the sowing distance between rows was 70 cm.

Sowing was executed in all systems with an MT-6 seeder (Seeding machinery Maschio Gaspardo MT 6R, Târgu-Mureș, Romania) on the same day; at the same time as sowing, basic fertilization was applied, which was later supplemented by additional fertilization during the phenophase of the maize with 6–7 leaves (intensive phase of absorption of fertilizers). Fertilization variants were established in relation to the requirements of the Green Deal project of the European Commission, which necessitates a reduction in chemical fertilizer doses [56].

Weed control was realized in two stages: in the pre-emergence stage using 0.4 L ha$^{-1}$ product with isoxaflutol 240 g L$^{-1}$ and cyprosulfamide (safener) 240 g L$^{-1}$ + 1.4 L ha$^{-1}$ based on dimethenamid-P (optically active) 720 g L$^{-1}$ (Frontier Forte, BASF, Kaiserslautern, Germany) and in the post-emergence stage, during the 3–5 leaf maize phenophase, using 1.0 L ha$^{-1}$ product based on fluroxypyr 250 g L$^{-1}$ (Tomigan 250 EC, Adama, Bucharest, Romania) for dicotyledonous weeds + 1.5 L ha$^{-1}$ based on 40 g L$^{-1}$ nicosulfuron (Nicogan 40 OD, Adama, Bucharest, Romania) for monocotyledonous weeds.

The maize was harvested with a Wintersteiger experimental plot combine (Wintersteiger AG, Ried im Innkreis, Austria), equipped with maize equipment, during the last 10 days of September and the first 10 days of October in all experimental variants. The grain yield obtained for each experimental variant was weighed and transformed using Romanian Standard Moisture into maize (RSM, 14% for maize).

### 2.6. Methods for Analysis and Processing of Experimental Data

The yield data were statistically processed using ANOVA [57], and the least significant difference (LSD, 5%, 1%, 0.1%) was established.

## 3. Results and Discussion

### 3.1. Climate Conditions during the Experimental Period

The thermal regime from April to September 2016–2022 is presented in Table 1. The data recorded at the Turda Meteorological Station [58] indicate an increase in monthly temperatures (Table 1), with visible warming of the weather for the entire vegetation period, starting from the emergence phase. The average monthly temperature of April has decreased in the last two years, as recorded by Șimon et al. [59], and is the only exception reported with regard to the average monthly growing season temperatures; the other temperatures exceeded the 65-year multiannual average, with deviations reaching up to 3–3.9 °C in the summer months when the reproductive organs are formed, greatly affecting the productivity elements. The recorded average temperatures of the six months (IV–IX) in almost all experimental years (except for the year 2021, where 16.5 °C was recorded) were above the multi-year average (16.3 °C). In 2018, the warmest period was recorded (an average of 18.8 °C). Maize plants that emerge under conditions where temperatures are lower immediately after emergence have more difficulties in developing, and, at very low temperatures, they fail to absorb nutritional elements, requiring a longer time to form each individual leaf [60]; this is why it is important that after emergence, the temperatures do not drop below 4 °C, a temperature at which the seedlings are affected by the cold, preventing their growth.

In the experimental area, there was an uneven distribution of the amount of precipitation between April and September (Table 2). Deviations from these average values were recorded, with the largest deviations being recorded during the flowering phase. Water stress is another key factor limiting yield, as it is dependent on rainfall conditions.

**Table 2.** Average monthly temperatures and rainfall for the period of April–September.

| Monthly Temperature (°C) | | | | | | | |
|---|---|---|---|---|---|---|---|
| Year/Month | IV | V | VI | VII | VIII | IX | Average IV–IX |
| 2016 | 12.4 | 14.3 | 19.8 | 20.5 | 19.6 | 17.1 | 17.3 |
| 2017 | 9.9 | 15.7 | 20.7 | 20.3 | 22.3 | 15.8 | 17.5 |
| 2018 | 15.3 | 18.7 | 19.4 | 20.4 | 22.3 | 16.7 | 18.8 |
| 2019 | 11.3 | 13.6 | 21.8 | 20.4 | 22.1 | 17.1 | 17.7 |
| 2020 | 10.3 | 13.7 | 19.1 | 20.2 | 21.5 | 17.8 | 17.1 |
| 2021 | 7.8 | 14.1 | 19.8 | 22.7 | 19.7 | 15.0 | 16.5 |
| 2022 | 8.8 | 16.3 | 21.1 | 23.1 | 22.3 | 14.3 | 17.7 |
| 65-year average | 10.0 | 15.0 | 18.1 | 19.9 | 19.5 | 15.2 | 16.3 |
| Monthly Rainfall (mm) | | | | | | | |
| Year/Month | IV | V | VI | VII | VIII | IX | Sum IV–IX |
| 2016 | 62.2 | 90.4 | 123.2 | 124.9 | 91.0 | 24.6 | 516.3 |
| 2017 | 65.2 | 65.4 | 30.6 | 110.2 | 36.1 | 56.2 | 363.7 |
| 2018 | 26.2 | 56.8 | 98.3 | 85.7 | 38.2 | 29.8 | 335.0 |
| 2019 | 62.6 | 152.4 | 68.8 | 35 | 63.8 | 19.4 | 402.0 |
| 2020 | 17.8 | 44.4 | 166.6 | 86.8 | 58 | 57.4 | 431.0 |
| 2021 | 38.4 | 80.8 | 45.0 | 123.1 | 52.9 | 39.1 | 379.3 |
| 2022 | 42.5 | 82.9 | 41.8 | 25.2 | 94.6 | 119.9 | 406.9 |
| 65-year average | 45.6 | 69.4 | 84.6 | 78.0 | 56.1 | 42.4 | 376.1 |

*3.2. Maize Emergence in Relation to the Experimental Factors*

In addition to climatic factors, the soil tillage system is of major importance for maize crops. The temperature and rainfall recorded after the sowing date affect the time period until emergence. In unconventional systems where the temperature of the soil is lower, a delay in the emergence of the crop by up to 1–4 days compared to the conventional tillage system has been observed. The lowest number of days from sowing (S) to emergence (E) was recorded in 2018 and 2019, being closely related to the higher temperatures recorded during the sowing period. Furthermore, in the last 3 years, due to the lower spring temperatures in 2021 and 2022 and the reduced precipitation in 2020, there was also a greater number of days between sowing and emergence, reaching up to 24 days (Table 3).

**Table 3.** Influence of soil tillage system and climatic conditions on crop emergence, 2016–2022.

| Year | 2016 | | 2017 | | 2018 | | 2019 | | 2020 | | 2021 | | 2022 | |
|---|---|---|---|---|---|---|---|---|---|---|---|---|---|---|
| Tillage System | S | E | S | E | S | E | S | E | S | E | S | E | S | E |
| CT | 19.04 | 03.05 | 21.04 | 05.05 | 04.05 | 15.05 | 16.04 | 27.04 | 15.04 | 04.05 | 23.04 | 07.05 | 03.04 | 25.04 |
| MTC | 19.04 | 03.05 | 21.04 | 05.05 | 04.05 | 15.05 | 16.04 | 28.04 | 15.04 | 04.05 | 23.04 | 07.05 | 03.04 | 20.04 |
| MTD | 19.04 | 02.05 | 21.04 | 07.05 | 04.05 | 16.05 | 16.04 | 26.04 | 15.04 | 06.05 | 23.04 | 08.05 | 03.04 | 21.04 |
| NT | 19.04 | 02.05 | 21.04 | 06.05 | 04.05 | 16.05 | 16.04 | 29.04 | 15.04 | 07.05 | 23.04 | 11.05 | 03.04 | 26.04 |
| No. days | 13–14 | | 15–17 | | 12–13 | | 12–14 | | 21–23 | | 15–19 | | 18–24 | |

Note: CT = conventional tillage; MTC = minimum tillage with a chisel; MTD = minimum tillage with a disk; NT = no-tillage; S = sowing data; E = emergence data; No. days = number of days from the sowing date until emergence.

*3.3. Maize Yields Obtained in Relation to the Experimental Factors*

Climatic changes, especially an increase in temperatures and a lack of precipitation, during the formation of reproductive organs are the main causes of significant production losses [61]. The adaptation of agricultural technologies is among the most accessible methods for reducing the impact of global warming [62]. Variations in environmental conditions during the growing season can have a major effect on crops, and factors such as water stress, heat, or a lack of nutrients during the growing season reduce yields [33,63,64].

After the statistical processing of the experimental data, it was found that the maize yield was significantly influenced by climatic conditions. Compared to the average yields achieved in the seven years (6429 kg ha$^{-1}$), the differences recorded in the other years showed inferior values in 2016, 2019, and 2022 and superior values in 2018, 2020, and 2021. The lowest yield was obtained in 2022 under the conditions of a prolonged drought and the heat of the summer period (Table 4). A low amount of precipitation, which correlates with high temperatures, causes plants to consume more energy to achieve significant increases in yield [65], with climatic conditions still being the most important factor determining crop yields [66].

**Table 4.** Maize yield obtained in 2016–2022 years.

| Experimental Factor | | Yield (kg ha$^{-1}$) | % | Difference, $\pm$ Control (kg ha$^{-1}$) |
|---|---|---|---|---|
| | $a_0$ year average | 6429 | 100 | control |
| A-Experimental year | $a_1$ 2016 | 6186 | 96 | −243 [00] |
| | $a_2$ 2017 | 6648 | 103 | 218 * |
| | $a_3$ 2018 | 6996 | 109 | 566 *** |
| | $a_4$ 2019 | 6202 | 97 | −228 [0] |
| | $a_5$ 2020 | 7354 | 114 | 925 *** |
| | $a_6$ 2021 | 6809 | 106 | 379 ** |
| | $a_7$ 2022 | 4811 | 75 | −1618 [000] |

LSD ($p$ 5%) = 158 kg ha$^{-1}$; LSD ($p$ 1%) = 240 kg ha$^{-1}$; LSD ($p$ 0.1%)= 389 kg ha$^{-1}$

Note: [000], *** = significant at 0.001% (negative and positive); [00], ** = significant at 0.01%; [0], *= significant at 0.05% (negative and positive).

For maize, temperatures exceeding 32 °C during the pollination period are a factor that causes significant yield losses. As Cairns [67] also states, heat stress alone or in combination with drought is a major constraint on maize yield. The amount of precipitation, especially its distribution throughout the growing season, is also of great importance in terms of yield because the maize crop in the Transylvanian Plain is dependent on precipitation [23], as precipitation and groundwater are the only sources of water available to plants throughout the year. In this study, the importance of the rainfall distribution is highlighted by the yield achieved. The yield results obtained during the period of 2016–2022 show that the highest yield was obtained in the years when the amount of water was more evenly distributed during the growing season, even if the annual amount was lower.

In the CT control variant, the yield achieved (7306 kg ha$^{-1}$) recorded a value close to that obtained in the MTC system (7231 kg ha$^{-1}$) and superior to that obtained in the MTD (6154 kg ha$^{-1}$) and NT variants (5026 kg ha$^{-1}$); these had a very significantly negative influence on the harvest, with the differences being between 1153 and 2280 kg ha$^{-1}$ (Table 5). The data obtained show that, under the soil conditions of Turda (with a clay texture), in terms of the depth of the tillage, for good development, the maize needs well-processed soil to develop its root system. Although some studies suggest that seeding in no-till or tilled land benefits yield, the results obtained in this study, as well as those obtained by Pittelkow et al. [68], argue that minimum tillage and no tillage should be combined with other conservation agricultural measures to exercise a positive effect on yield. In the research carried out by Wang et al. [69], it emerged that the no-tillage system had no significant effect on maize yield. However, following the research carried out by Liu et al. [70] in a semi-arid area of the Loess Plateau of China from 2014 to 2016, it was concluded that applying the no-tillage system increases the water capacity and bulk density of the soil, but soil porosity and maize yield are reduced.

**Table 5.** Influence of soil tillage system on maize yield, 2016–2022.

| Experimental Factor | | Yield (kg ha$^{-1}$) | % | Difference, $\pm$ Control (kg ha$^{-1}$) |
|---|---|---|---|---|
| B- Soil tillage system | $b_1$ CT | 7306 | 100 | control |
| | $b_2$ MTC | 7231 | 99 | −75 [ns] |
| | $b_3$ MTD | 6154 | 84 | −1153 [000] |
| | $b_4$ NT | 5026 | 67 | −2280 [000] |
| LSD ($p$ 5%) = 91 kg ha$^{-1}$; LSD ($p$ 1%) = 124 kg ha$^{-1}$; LSD ($p$ 0.1%) = 168 kg ha$^{-1}$ | | | | |

Note: CT = conventional tillage; MTC = minimum tillage with a chisel; MTD = minimum tillage with a disk; NT = no-tillage; ns = not significant; [000] = 0.001% *p*-value significant, negative values.

Maize is a nutrient-consuming plant, consuming nitrogen in particular, but the application of nitrogen at high doses can create an imbalance in the soil reaction [71], as high nitrogen fertilization creates a higher acidifying potential. The beneficial effects of additional fertilization with calcium ammonium nitrate (CAN) are better plant development and an increased yield. A difference of 374 kg ha$^{-1}$ compared to the variant with basic fertilization (control) presents a very significantly positive statistical result (Table 6). Moreover, other studies, including a study by Masood et al. [72], mention the effectiveness of CAN fertilizers in increasing maize yield.

**Table 6.** Influence of fertilization on maize yield, 2016–2022.

| Experimental Factor | | Yield (kg ha$^{-1}$) | % | Difference, $\pm$ Control (kg ha$^{-1}$) |
|---|---|---|---|---|
| C-Level of fertilization | $c_1$ $N_{56}P_{56}K_{56}$ | 6242 | 100 | control |
| | $c_2$ $N_{56}P_{56}K_{56}$ + $N_{40,5}$ $CaO_{12}$ | 6616 | 106 | 374 *** |
| LSD ($p$ 5%) = 55 kg ha$^{-1}$; LSD ($p$ 1%) = 74 kg ha$^{-1}$; LSD ($p$ 0.1%) = 99 kg ha$^{-1}$ | | | | |

Note: *** 0.001 *p*-value significant, positive values.

From the interaction of factors, namely experimental year x soil tillage system, yield differences were recorded annually between the four tillage systems and the CT control variant, with the variants of minimum tillage with a disk and no-tillage presenting very significantly negative statistical results. In the MTC variant, the differences are below 150 kg ha$^{-1}$ and do not present statistical significance. The yields obtained in CT and MTC throughout the research period have very similar values, which suggests the suitability of maize cultivation in the variant without plowing (Table 7).

**Table 7.** The influence of the interaction of experimental year $\times$ soil tillage system factors on maize yield, 2016–2022.

| Experimental Factor | | Yield (kg ha$^{-1}$) | % | Difference, $\pm$ Control (kg ha$^{-1}$) |
|---|---|---|---|---|
| 2016 | CT | 7063 | 100 | control |
| | MTC | 6914 | 98 | −149 [ns] |
| | MTD | 5970 | 85 | −1093 [000] |
| | NT | 4798 | 68 | −2264 [000] |
| 2017 | CT | 7922 | 100 | control |
| | MTC | 7668 | 97 | −255 [0] |
| | MTD | 6102 | 77 | −1820 [000] |
| | NT | 4899 | 62 | −3023 [000] |
| 2018 | CT | 8127 | 100 | control |
| | MTC | 8190 | 101 | 63 [ns] |
| | MTD | 6515 | 80 | −1612 [000] |
| | NT | 5150 | 63 | −2977 [000] |

**Table 7.** *Cont.*

| Experimental Factor | | Yield (kg ha$^{-1}$) | % | Difference, $\pm$ Control (kg ha$^{-1}$) |
|---|---|---|---|---|
| 2019 | CT | 6637 | 100 | control |
| | MTC | 6632 | 100 | $-5$ ns |
| | MTD | 6229 | 94 | $-408$ 00 |
| | NT | 5309 | 80 | $-1328$ 000 |
| 2020 | CT | 8481 | 100 | control |
| | MTC | 8384 | 99 | $-97$ ns |
| | MTD | 6796 | 80 | $-1686$ 000 |
| | NT | 5756 | 68 | $-2725$ 000 |
| 2021 | CT | 7599 | 100 | control |
| | MTC | 7619 | 100 | 20 ns |
| | MTD | 6888 | 91 | $-711$ 000 |
| | NT | 5128 | 68 | $-2471$ 000 |
| 2022 | CT | 5315 | 100 | control |
| | MTC | 5212 | 98 | $-103$ ns |
| | MTD | 4575 | 86 | $-741$ 000 |
| | NT | 4142 | 78 | $-1173$ 000 |
| LSD ($p$ 5%) = 242 kg ha$^{-1}$; LSD ($p$ 1%) = 329 kg ha$^{-1}$; LSD ($p$ 0.1%) = 444 kg ha$^{-1}$ | | | | |

Note: CT = conventional tillage; MTC = minimum tillage with a chisel; MTD = minimum tillage with a disk; NT = no-tillage; ns = not significant; 000 = 0.001% $p$-value significant; 00 = significant at 0.01%; 0 = significant at 0.05% (negative values).

Variations in temperatures and precipitation can have both positive and negative impacts on maize yield [73]. In the case of an increase in temperature during the vegetation period, which correlates with a reduced amount of rainfall, the maize yield is affected, even if other methods of reducing the impact are considered, such as the use of unconventional soil tillage systems or the changing of the sowing date. With several factors interacting and crop yields varying from year to year under the influence of meteorological factors, it is difficult to objectively evaluate the effectiveness of soil tillage methods [74] or other factors involved when using technology. Wang et al. [75] found that, although unconventional soil tillage systems do not play a significant role in improving yield, the benefits of the systems on soil health are still evident [76–78]. Optimal temperature ratios and water availability allow for the maximum yield to be obtained. After more than 30 years of observation, a strong positive correlation was revealed between corn production and the amount of precipitation, and a negative correlation was revealed with the sum of temperatures [79]. Thus, it can be stated that temperatures and precipitation during the vegetation period, which do not fall within the requirements of the crop, are the most important factors on which the yield of an agricultural crop depends [80–82].

Climatic conditions and technological factors are particularly important elements for any crop [62,83], but they are especially important for achieving a good maize yield [84]. Although the maize plant has a deep root system and can reach considerable soil depths, during the first phenophases, it still has high requirements for soil tillage. This statement can be clearly seen from the data presented in Table 8. The yields obtained in the MTD and NT systems in most years have distinctly significant or very significantly negative differences compared to those of the control. Regarding the MTC system, we can note that, except for in 2017, compared to CT, there are no statistically significant differences. Therefore, we can say that, under the climatic conditions of the Transylvanian Plain or under similar conditions, the MTC system could be a viable alternative to the CT system for maize crops. We must also bear in mind that the MTC system can also lead to a reduction in fuel consumption.



**Table 8.** The influence of the interaction of soil tillage system × experimental year × fertilization factors on maize yield, 2016–2022.

| Experimental Factor | | | Yield (kg ha$^{-1}$) | % | Difference, ± Control (kg ha$^{-1}$) | Duncan Test |
|---|---|---|---|---|---|---|
| B1 | | | 6779 | 100 | control | NML |
| B2 | A1 | C1 | 6633 | 98 | −146 $^{ns}$ | ONM |
| B3 | | | 5831 | 86 | −948 $^{000}$ | S |
| B4 | | | 4640 | 67 | −2139 $^{000}$ | X |
| B1 | | | 7347 | 100 | control | JI |
| B2 | A1 | C2 | 7195 | 98 | −152 $^{ns}$ | KJI |
| B3 | | | 6109 | 83 | −1238 $^{000}$ | SRQ |
| B4 | | | 4958 | 68 | −2389 $^{000}$ | XWU |
| B1 | | | 7708 | 100 | control | HG |
| B2 | A2 | C1 | 7367 | 96 | −341 $^{0}$ | JI |
| B3 | | | 5957 | 77 | −1751 $^{000}$ | SR |
| B4 | | | 4641 | 60 | −3067 $^{000}$ | X |
| B1 | | | 8137 | 100 | control | FEDC |
| B2 | A2 | C2 | 7969 | 98 | −169 $^{ns}$ | GFED |
| B3 | | | 6248 | 77 | −1890 $^{000}$ | RQP |
| B4 | | | 5157 | 63 | −2981 $^{000}$ | WVUT |
| B1 | | | 7884 | 100 | control | GFE |
| B2 | A3 | C1 | 7994 | 101 | 110 $^{ns}$ | GFED |
| B3 | | | 6404 | 81 | −1481 $^{000}$ | QPO |
| B4 | | | 4939 | 63 | −2946 $^{000}$ | XWU |
| B1 | | | 8370 | 100 | control | CB |
| B2 | A3 | C2 | 8387 | 100 | 17 $^{ns}$ | CBA |
| B3 | | | 6627 | 79 | −1743 $^{000}$ | ONM |
| B4 | | | 5362 | 64 | −3008 $^{000}$ | UT |
| B1 | | | 6497 | 100 | control | PONM |
| B2 | A4 | C1 | 6451 | 99 | −46 $^{ns}$ | PON |
| B3 | | | 6113 | 94 | −384 $^{0}$ | SRQ |
| B4 | | | 5160 | 79 | −1337 $^{000}$ | WVUT |
| B1 | | | 6777 | 100 | control | NML |
| B2 | A4 | C2 | 6813 | 101 | 36 $^{ns}$ | ML |
| B3 | | | 6346 | 94 | −432 $^{00}$ | QPO |
| B4 | | | 5458 | 81 | −1320 $^{000}$ | T |
| B1 | | | 8275 | 100 | control | DCB |
| B2 | A5 | C1 | 8192 | 99 | −83 $^{ns}$ | EDC |
| B3 | | | 6615 | 80 | −1660 $^{000}$ | ONM |
| B4 | | | 5494 | 66 | −2781 $^{000}$ | T |
| B1 | | | 8688 | 100 | control | A |
| B2 | A5 | C2 | 8577 | 98 | −112 $^{ns}$ | BA |
| B3 | | | 6976 | 80 | −1712 $^{000}$ | LK |
| B4 | | | 6018 | 69 | −2670 $^{000}$ | SR |
| B1 | | | 7375 | 100 | control | JI |
| B2 | A6 | C1 | 7499 | 102 | 124 $^{ns}$ | IH |
| B3 | | | 6608 | 90 | −767 $^{000}$ | ONM |
| B4 | | | 5062 | 67 | −2313 $^{000}$ | WUT |
| B1 | | | 7824 | 100 | control | GF |
| B2 | A6 | C2 | 7740 | 99 | −84 $^{ns}$ | HG |
| B3 | | | 7169 | 92 | −655 $^{000}$ | KJI |
| B4 | | | 5194 | 66 | −2630 $^{000}$ | WVUT |
| B1 | | | 5226 | 100 | control | VUT |
| B2 | A7 | C1 | 5216 | 100 | −10 $^{ns}$ | WVUT |
| B3 | | | 4269 | 82 | −957 $^{000}$ | Y |
| B4 | | | 3959 | 76 | −1267 $^{000}$ | Z |
| B1 | | | 5405 | 100 | control | T |
| B2 | A7 | C2 | 5209 | 96 | −197 $^{ns}$ | WVUT |
| B3 | | | 4881 | 90 | −524 $^{00}$ | XW |
| B4 | | | 4325 | 80 | −1080 $^{000}$ | Y |
| LSD ($p$ 5%) = 317.90 kg ha$^{-1}$; LSD ($p$ 1%) = 430.61 kg ha$^{-1}$; LSD ($p$ 0.1%) = 577.58 kg ha$^{-1}$ | | | | | | |

Note: ns = not significant; $^{000}$ = 0.001% $p$-value significant; $^{00}$ = significant at 0.01%; $^{0}$ = significant at 0.05% (negative values).

## 4. Conclusions

The experimental results obtained between 2016 and 2022 highlight the fact that under the conditions of chernozem soils with a high clay content in the Transylvanian Plain area, cultivating maize with superficial soil mobilization or no tillage using a Seeder Maschio Gaspardo MT 6R is not suitable, as maize requires deeper mobilization (and agricultural machines capable of doing this), and the yield data confirm this fact. The MTC system can be considered an alternative to CT, as the yield difference between the two systems is insignificant (below 100 kg ha$^{-1}$). Additional fertilization with calcium ammonium nitrate increases the yield by about 380 kg ha$^{-1}$. The success of the maize crop depends on the climatic conditions and the realization of quality sowing, with the soil moisture and the temperatures at sowing being particularly important due to their influence on the emergence and density of the crop and the distribution of rainfall from June to July in particular affects the yield of maize.

The measures recommended following this research, including for future research, are related to the introduction of new agricultural machines for minimum soil tillage and no-tillage, as well as other elements of agricultural technology to limit and counteract the effects of drought periods, being able to be adapted according to the particularities of the geographical area and pedoclimatic conditions. Such measures refer to the following: the use of a biological material that shows resistance to water and thermal stress; the use of agrotechnical measures favorable for the accumulation, conservation, and effective utilization of water from precipitation; the use of a conservative farming system based on protecting the soil and avoiding desertification; the identification of areas vulnerable to climate change; and the use of biological material that presents biological characteristics and pedoclimatic requirements specific to the new climatic trends of areas vulnerable to climate risks.

**Author Contributions:** Conceptualization, F.C. and C.C.; methodology, T.R.; software, A.Ș.; validation, A.O.C. and F.C.; formal analysis, T.R.; investigation, M.B. and A.T.; resources, A.-M.V.; data curation, A.-M.V.; writing—original draft preparation, F.C.; writing—review and editing, T.R.; visualization, F.C.; supervision, T.R.; project administration, F.C.; funding acquisition, C.C. All authors have read and agreed to the published version of the manuscript.

**Funding:** This work was supported by the Ministry of Agriculture and Rural Development, by project ADER 20.1.3, Contract no. 20.1.3/2023, from the Sectoral Plan 2023–2026.

**Institutional Review Board Statement:** Not applicable.

**Informed Consent Statement:** Not applicable.

**Data Availability Statement:** Data are contained within the article.

**Conflicts of Interest:** The authors declare no conflict of interest.

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
