# Peer review of "Application of Unconventional Tillage Systems to Maize Cultivation and Measures for Rational Use of Agricultural Lands"

_land, doi:10.3390/land12112046_

Round 1

Reviewer 1 Report

Comments and Suggestions for Authors

For a clear presentation of the importance of research for corn crop systems in the studied area, but also for a correct interpretation of the results obtained, it is recommended to review/correct/reformulate as necessary the following lines/paragraphs of the manuscript:

Introduction

36 problems for maize

37 deficient thermal regime - contrary to the statements of 173, 174, 175, 176, 177 - The amount of precipitation, especially its distribution throughout the growing season, is also of great importance in the yield, because the maize culture in the Transylvanian Plain is dependent on precipitation [23], as it and groundwater are the only sources of water available to plants throughout the year

42 water protection

2. Materials and Methods

67 Esperimental factors - experimental

80 two repetitions?

109 Climatic conditions? - only data relating to average monthly temperatures are presented

126 Reformulate the title of the table in accordance with the data presented - average monthly temperatures

112,113 – The average monthly temperature of April and May has registered a decrease in the last three years Table 1. April 2020 -10.3(ºC)/ 65 years average 10.0(ºC), May 2022 – 16.3(ºC) / 65 years average 15.0(ºC)

131 - the pre- and post-bloom – recommended – flowering phase (phenophase)

4. Discussion

158 the adaptation of agricultural technologies being among the most accessible methods of reducing the impact of global warming [35], analyze the statement in relation to the author's research on rice (Ali, S.; Ghosh, B.C.; Osmani, A.G.; Hossain, E. Fogarassy, ​​C. Farmers' Climate Change Adaptation Strategies for Reducing the Risk of Rice Production: Evidence from Rajshahi District in Bangladesh. Agronomy 2021, 11, 600. https://doi.org/10.3390/agronomy11030600)? 182 Table 4. Influence of the year on maize yield, 2016-2022 - Difference ± control (kgha-1 ) 189 ....maize is demanding... 195, 208,235 - (kgha-1 ) 203,204 – explanation of the beneficial effect of calcium ammonium nitrate (CAN) in the conditions of the research carried out 219 - Changes in temperatures and precipitation – reformulated 221- a poor rainfall background – reformulated 246-249 Reformulation according to the factors studied, the data recorded and the results obtained. It would also be important to analyze the interaction of the studied factors (AXBXC), respectively the DUNCAN test. The results of the research could outline technological and economic solutions favorable to the farmer and the environment

Author Response

Dear Reviewer,
Thank you very much for your constructive recommendations. These helped us to improve the paper, to increase the accuracy and reproducibility of the experiment presented in the paper.
All the recommendations made by you have been implemented and marked in the paper with red color.
Any other recommendation to improve the paper is welcome and we will gladly do it.
All the best

Reviewer 2 Report

Comments and Suggestions for Authors

Dear Authors,

Thank you for submitting your manuscript to Land. I believe you have put in a lot of effort into the article; however, I cannot recommend it for acceptance at this stage. I have a few recommendations that might improve the quality of your manuscript:

Abstract:

What is "cocini"?

Why did you not mention here the two variants of fertilization?

Keywords: Please do not repeat words from the title of the manuscript. "Pedo-climatic conditions" seem inappropriate in this manuscript.

Introduction:

This chapter is too general.

The introduction should be significantly enhanced.

There needs to be more comparison of different tillage methods (depending on soil type) and their effects on maize.

There needs to be add research hypotheses.

Materials and Methods:

Correct and unify units throughout the chapter.

For the "MT-6 seeder," please specify the brand.

Did the maize always grow on the same plot for the whole duration of the experiment?

What is "STAS moisture"?

I recommend creating a table in this chapter that includes all the operations on the plot and their dates, as well as a table for soil properties.

Please be more specific about yield evaluation and emergence.

Results:

Weather is not a result. Please merge tables 1 + 2, and in addition, I suggest moving this information to the previous chapter.

Discussion:

The difference between CS and MTC is one percent. Can you recommend MTC instead of CS? How do these methods compare in terms of fuel consumption?

Was fertilization done for all types of tillage?

Conclusion:

There is a lack of practical recommendations.

References:

Please edit them into the correct format.

Fingers crossed for the manuscript correction!

Comments on the Quality of English Language

Proofreading recommended.

Author Response

(The authors gave the same response as above.)

Reviewer 3 Report

Comments and Suggestions for Authors

General comments:

1.       On title: The Application of Unconventional Tillage Systems to Maize Cultivation, Measure for Rational Uses of Agricultural Lands. You may remove “The”. also, if the “measure of for rational uses of agricultural lands” is also a result of unconventional tillage system – insert “and” before measure for…

2.       On abstract: Abstract is being published separately, consider removing acronyms in the abstract. You may first mention the acronyms in the Introduction part.

3.       The introduction did not mention the objectives. Objectives of the study should be written in the latter part of the introduction.

4.       In the Introduction. You may discuss the economic importance of maize, like the data on domestic consumption, is it being exported, etc.

5.       In the Introduction, discuss the previous study on the soil type (chernozem soils).

6.       On Materials and Methods: Add a subsection for “Experimental Site” and write a description of the location, population, general weather or climate, water supply, rainfall, etc.

7.       Insert a Figure (a map) showing the experimental site.

8.       Section 2.2 Research Method may be replaced by Section 2.2 Experimental Design and rewrite the whole section by establishing the effect that a factor or independent variable has on a dependent variable.

9.       Add a section in Materials and Methods on the “Collection or Recording of Weather and Rainfall Data”.

10.   Insert some important Figures like the diagram of planting set up, or photo of maize grown, or the photo of research station where the study conducted.

11.   Merge the “Results” and “Discussion” sections. As the current manuscript, the content of “discussion” does not discuss the content of “results”.

Author Response

(The authors gave the same response as above.)

Round 2

Reviewer 2 Report

Comments and Suggestions for Authors

Dear Authors,

thank you for revising the manuscript. At this stage, I recommend accepting your article, but English proofreading is necessary.

Comments on the Quality of English Language

English proofreading is necessary!